Accepted at the ICLR 2024 Workshop on AI4Differential Equations In Science

# NEURAL ODE FOR MULTI-CHANNEL ATTRIBUTION

**Yudi Zhan, Oshry Ben-Harush, Siyu Zhu, Xin Liang, Te Ken, Daryl Hammett**
Amazon Web Services, Seattle, WA, USA
`{yudzhang, oshryb, zhusiyu, liangxin, teken, darylham}@amazon.com`

## ABSTRACT

Multi-Touch Attribution plays a crucial role in both marketing and advertising, offering insight into the complex series of interactions within customer journeys during transactions or impressions. This holistic approach empowers marketers to strategically allocate attribution credits for conversions across diverse channels, not only optimizing campaigns but also elevating overall marketplace strategies. In this paper, we acknowledge the irregular time series nature of customer journey data and showcase both the effectiveness and limitations of neural ordinary differential equations in terms of estimating the attributions and predicting conversions.

## 1 INTRODUCTION

With the rapid growth of internet users over the past few years, companies keep tracks of their customers behaviors through multiple marketing channels, for example, paid search, natural search, banner ads. It is crucial for the companies to attribute the right conversion credit on the interaction between the user and the website content along the customer journey (touchpoints). Ideally, such analysis should be done through A/B testing. However, usually the companies cannot perform a mass-scale A/B test, as that could have negative impact on customer experience on important websites, and usually requires spending a huge budget. Since budget allocation over different channels is always an important decision, it is crucial for them to use multi-touch attribution (MTA) methods to estimate the relative contribution of each touchpoint in customer conversion journeys.

In the past, rule-based methods like first-touch and last-touch attributed all credit to the initial/final touchpoint. Later, MTA models such as Linear, Time Decay, and U-Shaped emerged, distributing credit across touchpoints based on perceived influence. However, these models oversimplify real-world complexities, struggling to capture nuanced interaction impacts. The Sharpley value, rooted in cooperative game theory (Zhao et al. (2018); Singal et al. (2019); Shao & Li (2011)), captures the marginal contribution of each touchpoint. Drawbacks include a failure to capture sequential dependencies and factorial calculations due to considering all touchpoint permutations. The Markov Chain model, combined with the removal effect (Anderl et al. (2016); Archak et al. (2010)), uses transition probabilities between states/touchpoints in a customer's journey. However, higher-order Markov Chains are computationally intensive. A causal motivated methodology involves survival analysis, focusing on the conversion event's predictive goal (Ren et al. (2018)). Dalessandro et al. (2012) proposed assigning conversion credits using a causal cooperation model, addressing practical constraints and proposing an approximate attribution measure. However, a drawback of these models lies in the absence of exogenous variation in user historical data, posing a risk to the reliability of attribution results due to bias in the counterfactual predictor's training data caused by confounders, making accurate predictions more challenging. Inspired by methods proposed to address such problem in longitudinal data, Yao et al. (2022) proposed a CausalMTA method which eliminates the confounding bias from both static and dynamic perspectives and learn an unbiased conversion prediction model using historical data.

Nevertheless, these methodologies lean towards single-point prediction and overlook sequential patterns in user browsing history. Attribution credits derived from these methods often rely on heuristic additive assumptions, which may prove ineffective in practical scenarios. Additionally, assumptions about survival function, such as exponential hazard function or Weibull distribution, constrain the model's capacity to adapt to diverse real-world data. Moreover, such customer journey are complex and presents non-uniform intervals, which is a common challenge to models using conventional recurrent neural networks (RNNs) or temporal convolutional neural networks (TCNs) (Chen

et al. (2019), Rubanova et al. (2019)). To address this limitation, Rubanova et al. (2019) extended RNNs by incorporating continuous-time hidden dynamics defined by ordinary differential equations (ODEs), introducing a model known as ODE-RNN. These models exhibit a natural ability to handle arbitrary time gaps between observations and explicitly model the probability of observation times through Poisson processes.

This paper introduces a novel application of ODE-LSTM (Lechner & Hasani (2020)) to the MTA problem, where we attempt to estimate attribution within customer journeys by incorporating an attention mechanism into the original model. We conduct a comparative evaluation of this approach alongside other commonly employed MTA methods, revealing both its strengths and weaknesses. Our findings suggest that ODE-LSTM outperforms other methods particularly in scenarios where time intervals are not excessively irregular, albeit demonstrating a decline in performance as irregularity increases. However, it excels in estimating attributions compared to alternative approaches.

## 2 METHODOLOGY

Let $\mathcal{X} = (\mathbf{X}_1, \mathbf{X}_2, \ldots, \mathbf{X}_N)$ denote the input sequence data, where each $\mathbf{X}_i = (\mathbf{x}_1, \mathbf{x}_2, \ldots, \mathbf{x}_L)$ represents a sequence with length $L$ and $\mathbf{x}_l$ denotes the features at the $l$th location. Let $\mathcal{Y} = (y_1, y_2, \ldots, y_N)$ be the class of the sequential data.

### 2.1 ODE-LSTM WITH ATTENTION

Follow Chen et al. (2019) and Lechner & Hasani (2020), we use autoregressive modeling with ODE-LSTM with an additional attention layer to model the customer journey seuqences. Assume each input data $\mathbf{x}_l$ is associated with an timestamp $t_l$ and denote hidden states as well as memory cell as $\mathbf{h}_l$ and $\mathbf{m}_l$. The ODE-LSTM algorithm follows:

$$\mathbf{h}_l', \mathbf{m}_l = \mathrm{LSTMCell}\left(\mathbf{m}_{l-1}, \mathbf{h}_{l-1}, \mathbf{x}_l\right),$$

$$\mathbf{h}_l = \mathrm{ODESolve}\left(f_\theta, \mathbf{h}_{l-1}, \mathbf{h}_l', (t_{l-1}, t_l)\right),$$

where the function $f_\theta$ specifies the dynamics of the hidden state, using a neural network with parameters $\theta$.

In order to obtain the attribution, the above hidden states are further feed to an attention layer to identify pivotal touchpoints contributing to conversions. Subsequently, we consolidate the representations of these significant touchpoints, creating a comprehensive context vector.

$$
\begin{aligned}
\mathbf{v}_l &= \tanh\left(\mathbf{W}\mathbf{h}_l\right) \\
a_l &= \frac{\exp\left(\mathbf{v}_l^T \mathbf{u}\right)}{\sum_l \exp\left(\mathbf{v}_l^T \mathbf{u}\right)} \\
\mathbf{s} &= \sum_l a_l \mathbf{h}_l
\end{aligned}
\tag{1}
$$

The hidden states $\mathbf{h}_l$ are feed through a one-layer multilayer perceptron (MLP) to get $\mathbf{v}_l$, where $\mathbf{W}$ is a learnable matrix. Then, we measure the importance of the touchpoint by assessing the similarity of $\mathbf{v}_l$ with the vector $\mathbf{u}$ and obtain a normalized importance weight $a_l$ through a softmax function. It is noteworthy that, by design, $a_l > 0$. This construction offers the advantage that the contribution of every touchpoint is always positive. Afterward, we compute the vector $\mathbf{s}$ as the weighted sum of touchpoint representations based on the non-negative weights. Essentially, $\mathbf{s}$ is the convex combination of all $\mathbf{h}_l$. $\mathbf{u}$ can be seen as a high-level representation of a fixed sequence. We can customize this attribution model by imposing constraints on $\mathbf{u}$ based on domain knowledge about touchpoint importance, it can either be kept fixed or initialized randomly and jointly learned during the process. In our modeling, we adopt the latter approach. See Appendix A.1 for model structure visualization.

In MTA problem, customer journeys are categorized into positive (leading to conversions) and negative (not leading to conversions). This problem can be treated as binary classification in the transformed journey vector space $\mathbf{s}$, which combines hidden outputs and attention weights. Thus, we optimize a cross-entropy loss to train the model.

## 2.2 ATTRIBUTION CALCULATION AND VALIDATION

Up to this point, we have the estimated conversion probability $p(y|\mathbf{X}_i)$ and the attention scores $\mathbf{a}$. With these outcomes, we can inherently allocate attribution to channels at each touchpoint $l$. As we want to estimate the impact of each channel on successful conversions, our calculations exclusively focus on customers who have achieved successful conversions. The total attribution of a channel is the accumulative sum of the touchpoint attention scores if that touchpoint visit that channel.

Overall, we have two directions to measure the model performance. The first one focuses on conversion estimation performance, where we use AUC and PRAUC. The second part aims at the performance of calculated attributions for various channels. We evaluate attribution consistency across channels using the Jaccard Index. Additionally, we introduce a novel metric called AURE (Area Under Removal Effects). AURE tracks the cumulative impact on conversion probabilities by successively removing channels based on the ranked attributions. See Appendix A.2 for details.

## 3 EXPERIMENTS

### 3.1 COMPETING METHODS

We have three methods to compare **Transformers (TRANS)** (Vaswani et al. (2017)), **Temporal Convolutional Neural Networks (TCN)**, and **Attention LSTM (ALSTM)**. For Transformers and TCN, Integrated Gradients (Sundararajan et al. (2017)) are utilized to explain the importance of original input features. For ALSTM, after LSTM layers, we added the same attention layer as introduced in 1 to obtain the attribution.

### 3.2 DATASETS

We conduct our experiments on two real-world datasets. See Appendix A.3 for experiment details.

**Criteo**, is a dataset of real-time auction-based advertising attribution modeling (Diemert et al. (2017)). This dataset captures the dynamics of Criteo's live traffic over an extensive 30-day period. It encompasses over 16 million impressions and documents 45,000 conversions across 700 campaigns. Each impression record potentially corresponds to click actions, with labeled touch points indicating whether a click occurred. The dataset includes 12 features in total, including the time, and the max customer journey length is 20. The time range is normalized to 0 to 1, and the time differences can be nearly 0 or as large as 0.9.

**Marketing Sign-up Data**, which contains the customer actions over one year span, each timestamp corresponds to a user's visit to a particular URL. For every visit, we record four numeric features and eight categorical features. The data has about 1.4 million visitors, over 7.6 million visits and about 9,000 URLs for attribution estimation. We limit the max customer journey length as 15 timestamps in this data. Since the time span is over a year, the timestamps gaps could be one month or a few seconds. We use days as timestamp in this data, in this case, most of the time differences are 0, however, there exists large time difference such as 30 (days).

### 3.3 PERFORMANCE

#### 3.3.1 CONVERSION ESTIMATION PERFORMANCE

| Models | Criteo | | Marketing Sign-up | |
|---|---|---|---|---|
| | AUC | PRAUC | AUC | PRAUC |
| ODE-LSTM | **0.9832** | **0.9293** | 0.9200 | 0.8507 |
| ALSTM | 0.9827 | 0.9286 | **0.9710** | **0.9292** |
| TCN | 0.9817 | 0.9238 | 0.9629 | 0.9079 |
| TRANS | 0.9813 | 0.9226 | 0.9262 | 0.8394 |

Table 1: Performance Metrics for Criteo and Marketing Sign-up datasets.

Table 1 demonstrates the model performance in terms of the AUC and PRAUC. On the data (Criteo) that the time scale are relative small and most time differences are valid, ODE-LSTM performs

the best, however, as the time difference goes large and most differences are 0 as in the marketing signup data, ODE-LSTM was beat by ALSTM and TCN. The reason might be the ODE-LSTM faces challenges due to its focus on continuous transitions. ALSTM and TCN are more robust in capturing patterns in such scenarios. However, the simple Transformer model is the worst on the both data. It might be because its lack of inherent temporal understanding, which means it may not capture important temporal dependencies in the data. For such irregular time intervals or missing data points data, Transformers may not handle such irregularities well, and additional preprocessing or better time encoding is often needed.

### 3.3.2 ATTRIBUTION ESTIMATION

We plot AURE curves to validate attributions in Figure 2. It appears that ODE-LSTM and Transformers are highly consistent on Criteo data. The top 100 removal effects for ODE-LSTM and Transformer are 0.801 and 0.800 respectively, indicating a slight better performance of ODE-LSTM. The differences among methods on the marketing data is more significant (Appendix Table 2 also demonstrates this), although ODE-LSTM is not the best model in terms of AUC, the top 100 removal effects is the highest as 0.888, and it has a dominating trend in Figure 2b. Additionally, to compare the actual scores, Figures 1 show the normalized attribution scores of selected ten channels.

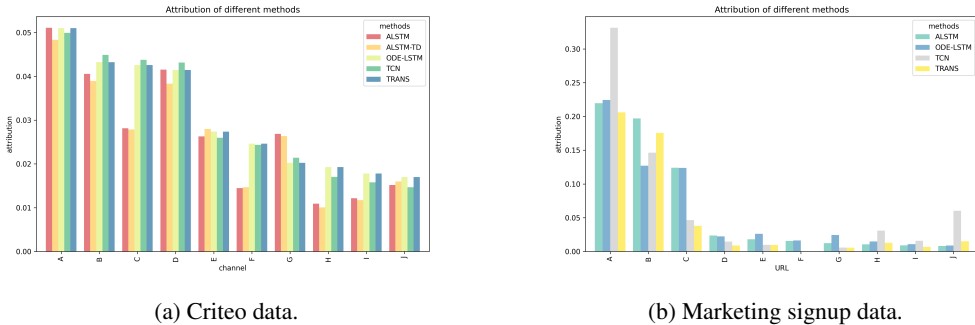

|     (a) Criteo data.     |     (b) Marketing signup data.     |

Figure 1: Attribution comparisons on top 10 channels selected by the best performance model given each data.

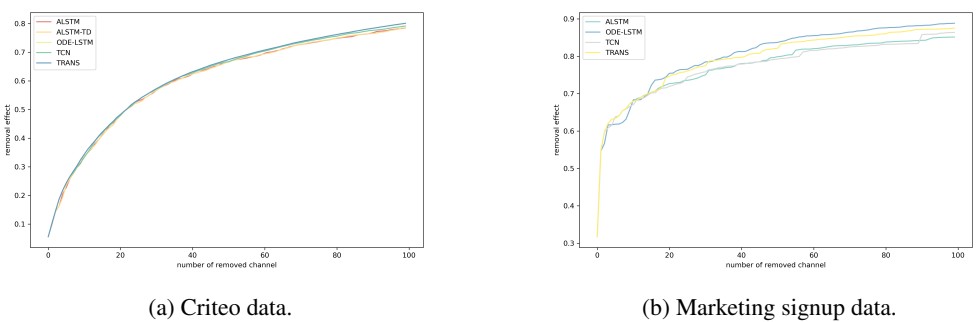

|     (a) Criteo data.     |     (b) Marketing signup data.     |

Figure 2: AURE on top 100 channels selected by each method.

## 4 DISCUSSION

We proposed a ODE-LSTM combined with an attention mechanism to estimate the attribution in MTA problem. ODE-LSTM is not necessary the best the model if simply comparing the AUC and PRAUC, and ALSTM is the most robust method for predicting conversion. However, by comparing the proposed AURE metrics, ODE-LSTM gives the best results. Although Neural ODE handles continuous data, e.g. irregularly-sampled data or test-time sampling shift automatically and are mathematically tractable to analyze. They are extremely slow at both training and inference. To

further enhance the performance of ODE-related methods, a potential avenue for improvement lies in refining the handling of time dynamics inherent in the attribution problem. Or we could try newly developed methods, for example NCDE (Kidger et al. (2020)) and State Space Model (Gu et al. (2022); Gu & Dao (2023)).

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

## A  APPENDIX

### A.1  ATTENTION FOR ATTRIBUTION

Below Figure 3 is the attention module we used to obtain the attribution $a_l$, there are some potential extension such that we can consider the time decay effects. Each touchpoint is associated with its occurrence time. The time difference between the occurrence time and the end time can be considered as a factor, smaller indicates that the timestamp is closer to the end time, potentially has more attribution to the final conversion or non conversion state, we can penalize the attention scores by this time decay factor. Moreover, the vector $\mathbf{u}$ can be defined as a representation of conversion or non conversion state, which makes attribution closer to capturing the label information.

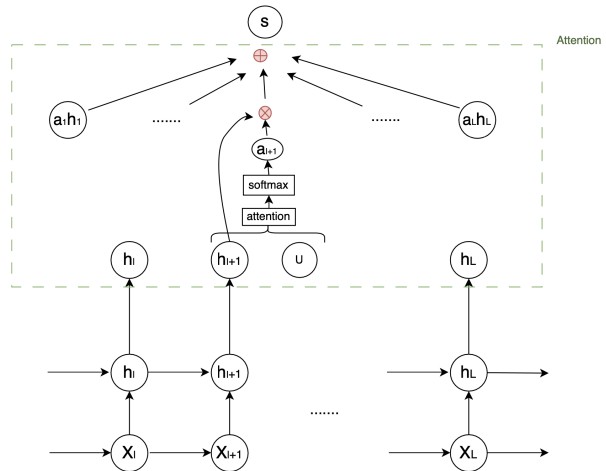

Figure 3: Attention for Attribution

### A.2  AURE

Assume the attributions accurately reflect a channel's influence on conversion, eliminating highly influential channels should lead to a substantial decline in conversion probability. By plotting a curve with the x-axis representing the count of removed channels starting from the highest attribution, and the y-axis indicating the difference between the initially trained best probability and the probability after removal, we can consider the curve's area under it as a statistical metrics, the larger area signifies a better method.

Let $c_1, c_2, \ldots, c_m$ represent channels, $P(y|\mathbf{X}_i)$ represent the conversion probability for instance $i$. The removal effect for instance $i$ after removing channel $\tilde{c}_j$ is given by:

$$R_{ij} = 1 - P(y_i|\mathbf{X}_{i,\tilde{c}_j})/P(y_i|\mathbf{X}_i),$$

where $\mathbf{X}_{i,\tilde{c}_j}$ is denoted as removing the channel $\tilde{c}_j$ in instance $i$.

For plotting the curve, the x-axis represents the number of channels removed, starting from the highest attribution: $j = 1, 2, \ldots, m$. The y-axis indicates the expected removal effect:

$$\frac{1}{N}\sum_{i=1}^{N} R_{ij}.$$

## A.3 TRAINING DETAILS

Both datasets are divided into trainset and testset (ratio: train 0.8, test 0.2). For Criteo data, 20% is positive labelled, and the detailed features includes time, click, campaign, and additional nine categorical features. There are about 700 channels(campaigns) to evaluete and estimate the attribution.

For Marketing data, 30% is positive labelled. And the features include channel, visited URLs, and other categorical attributes, such as time, page view duration, scroll depth, and other relevant metrics.

For the hyperparameters, we set epoch as 20, learning rate as $1^e-4$, weight deacay as $1^e-5$, hidden dimension as 128 and the dropout rate is 0.2. LSTM layers, attention heads and Temporal convolutional layers are all set as 2.

## A.4 TABLES AND FIGURES

|  | ODE-LSTM | ALSTM | TCN | TRANS |
|---|---|---|---|---|
| ODE-LSTM |  | 0.60 | 0.316 | 0.526 |
| ALSTM | 0.7094 |  | 0.324 | 0.550 |
| TCN | 0.754 | 0.7857 |  | 0.48 |
| TRANS | 0.9417 | 0.6949 | 0.7857 |  |

Table 2: Comparison of Jaccard Index of top 100 ranked channels given by each method. The green color represents Jaccard Index of Marketing signup data, and the blue color belongs to Criteo data.

