# OpenReview forum: "Neural ODE for Multi-channel Attribution"
_ICLR.cc/2024/Workshop/AI4DiffEqtnsInSci — AI4DiffEqtnsInSci @ ICLR 2024 Poster_

### Official Review · Reviewer_A1VX · 2024-02-23
**Review for Neural ODE for Multi-channel Attribution**

**Rating:** 7
**Confidence:** 3

**Review:**

This paper investigates using a neural ODE-LSTM with an attention mechanism to carry out the task of multi-touch attribution. They compare its performance to a transformer, temporal CNN and an attention LSTM, using two different datasets (auction-based advertising and marketing sign up data). They find varying performance between these models depending on the test metric and test dataset used, and discuss the limitations/advantages of each model.

In general, the paper seems to offer a novel approach for solving the MTA problem, and it appears to be competitive with the other benchmarks tested. Furthermore, the paper is concise, well-written, and appears robust in its ML workflow and tests.

I see a few weaknesses that could be addressed:

-	Is there any justification why an ODE should be used to model this data, apart from the fact it is more amenable to irregular time-series? Do we expect customer behavior to be determined by an ODE? Given that the other models tested perform in the same ball-park, I am unsure if this data is really generated from an underlying ODE.

-	The authors mention other simpler MTA models in the introduction (Linear, Time Decay, U-Shaped models etc). If possible, it would be useful to compare against these as baseline approaches to see how much better the learned approaches perform

-	Section 2.1 – typo in the word ‘sequences’

---

### Official Review · Reviewer_xEvN · 2024-02-27
**Informative application of ODE-LSTM to estimate the contribution of each touchpoint in a customer’s journey to a conversion event**

**Rating:** 6
**Confidence:** 2

**Review:**

This paper introduces a novel application of ODE-LSTM with an attention mechanism to the multi-touch attribution (MTA) problem, where the goal is to estimate the contribution of each touchpoint in a customer’s journey to a conversion event. The authors compare their approach with three competing methods on two real-world datasets and evaluate the performance in terms of conversion prediction and attribution estimation.

## Pros
1. ODE-LSTM can handle arbitrary time gaps between observations and model the probability of observation times through Poisson processes. (Section 1, Introduction)
2. ODE-LSTM outperforms other methods in scenarios where time intervals are not excessively irregular, and excels in estimating attributions compared to alternative approaches. (Section 3.3, Performance)
3. ODE-LSTM is highly consistent with Transformers in terms of attribution results on Criteo data, and has a dominating trend on Marketing sign-up data. (Section 3.3.2, Attribution Estimation)

## Cons
1. ODE-LSTM is not the best model in terms of AUC and PRAUC, and ALSTM and TCN are more robust in capturing patterns in scenarios with large and irregular time differences. (Section 3.3.1, Conversion Estimation Performance)
2. ODE-LSTM is extremely slow at both training and inference, due to its focus on continuous transitions. (Section 4, Discussion). This thing has been also pointed out in section 3.3.1. Some discussions on solutions to overcome this limitation would have been greatly welcome.
3. The authors use two real-world datasets, Criteo and Marketing Sign-up, but they are both related to online advertising and attribution modeling. They do not test their method on other domains or applications that could benefit from ODE-LSTM

---

### Meta-Review · Area_Chair_xNmk · 2024-03-01

**Recommendation:** Accept (Poster)

**Metareview:**

Authors propose an ODE-LSTM model to solve Multi-Touch Attribution (MTA) challenges. According to reviews, I suggest further justifying the motivation of using ODEs, addressing the speed limitations, and potentially testing other application domains beyond marketing. I recommend authors addressing concerns raised by the reviewers to prepare the paper for the camera-ready version.

---

### Decision · Program_Chairs · 2024-03-02

Accept (Poster)